ecology

functional diversity, ecological trait, fisheries, global change, ecosystem functioning, conservation

**Author for correspondence:**
Arnaud Auber
e-mail: arnaud.auber@ifremer.fr

# Rebound in functional distinctiveness following warming and reduced fishing in the North Sea

Juliette Murgier[1], Matthew McLean[2], Anthony Maire[3], David Mouillot[4], Nicolas Loiseau[4], François Munoz[5], Cyrille Violle[6] and Arnaud Auber[1]

[1]IFREMER, Unité Halieutique Manche Mer du Nord, Laboratoire Ressources Halieutiques, 150 quai Gambetta, BP699, 62321 Boulogne-sur-Mer, France
[2]Department of Biology, Dalhousie University, Halifax, Nova Scotia B3H 4R2, Canada
[3]EDF R&D LNHE - Laboratoire National d'Hydraulique et Environnement, 6 quai Watier, 78401 Chatou, France
[4]MARBEC, Univ. Montpellier, CNRS, IFREMER, IRD, 34095 Montpellier Cedex, France
[5]University Grenoble-Alpes, LIPHY, 38041 Grenoble Cedex 9, France
[6]CEFE, Univ. Montpellier, CNRS, EPHE, IRD, Univ. Paul Valéry Montpellier 3, Montpellier, France

MM, 0000-0001-6518-6043; AM, 0000-0003-0920-773X; DM, 0000-0003-0402-2605; NL, 0000-0002-2469-1980; FM, 0000-0001-8776-4705; CV, 0000-0002-2471-9226; AA, 0000-0002-8415-1652

Functionally distinct species (i.e. species with unique trait combinations in the community) can support important ecological roles and contribute disproportionately to ecosystem functioning. Yet, how functionally distinct species have responded to recent climate change and human exploitation has been widely overlooked. Here, using ecological traits and long-term fish data in the North Sea, we identified functionally distinct and functionally common species, and evaluated their spatial and temporal dynamics in relation to environmental variables and fishing pressure. Functionally distinct species were characterized by late sexual maturity, few, large offspring, and high parental care, many being sharks and skates that play critical roles in structuring food webs. Both functionally distinct and functionally common species increased in abundance as ocean temperatures warmed and fishing pressure decreased over the last three decades; however, functionally distinct species increased throughout the North Sea, but primarily in southern North Sea where fishing was historically most intense, indicating a rebound following fleet decommissioning and reduced harvesting. Yet, some of the most functionally distinct species are currently listed as threatened by the IUCN and considered highly vulnerable to fishing pressure. Alarmingly these species have not rebounded. This work highlights the relevance and potential of integrating functional distinctiveness into ecosystem management and conservation prioritization.

## 1. Introduction

Trait-based approaches are rapidly improving our understanding of how accelerating global change and biodiversity loss are impacting ecosystem multifunctionality [1,2]. Traits correspond to species ecological characteristics, such as feeding, reproductive or life-history strategies, which are linked to ecosystem functioning and mediate environmental responses [3–5]. Species differ in their combinations of traits, and losing species with very different traits compared to the rest of the community, i.e. functionally distinct species (*sensu* Violle *et al.* [6]), can have a major impact on ecosystem functioning if no other species can replace the potentially lost functions [6,7]. For instance, species highly specialized in resource use, which are often the most vulnerable to environmental variation and human pressure [8,9], support unique roles that may be performed by few, if any, other species in ecosystems [10]. Recent studies show that functionally distinct species disproportionately contribute to

community stability [11] and ecosystem functioning [12], and can support important ecosystem services [13]. These studies suggest that functionally distinct species should be considered as priority conservation targets to ensure sustainable ecosystem functioning and services [14].

The vulnerability, resistance and resilience of marine ecosystems to global change are of major concern as they support an array of essential services [15–17]. Global fisheries provide livelihoods for over 10% of the world's population [18] and sustaining fish stocks under increased demand due to global population growth is a major challenge [19]. Fish communities are key to ecosystem functioning, whether considered from a taxonomic, phylogenetic or functional point of view [20,21]. Along with climate change, overexploitation of fish stocks directly impacts marine biodiversity and can weaken the resilience of marine ecosystems [22,23]. Therefore, fisheries management requires a better knowledge of fish community responses, particularly in terms of species traits, to various anthropogenic pressures and environmental changes [24].

The North Sea, as a global warming 'hotspot' (with a 1.6°C rise in sea surface temperature observed over 25 years [25,26]) and overfished region [27], provides a unique opportunity to examine fish community responses to combined pressures of climate change and fishing. Ever increasing seawater temperature has led to poleward shifts in species distributions [28–30], and shifts toward deeper, colder environments [25,31]. Recent studies have documented changes in the taxonomic and functional structure of fish communities in the North Sea [2,32–34], which have altered their functioning with potential consequences for fisheries resources. In addition, the loss of species with unique features within fish communities could impact ecosystem multifunctionality. Yet, how functionally distinct species have responded to climate change and fishing in the North Sea has not been investigated.

Using 33 years of scientific surveys, we assessed the spatio-temporal dynamics of functional distinctiveness in fish communities of the North Sea during a period of climate warming and reduced fisheries exploitation. The objectives of this study were (i) to identify functionally distinct fish species and their ecological characteristics, (ii) to assess the combined effects of environmental drivers and fishing pressure on functional distinctiveness, and (iii) to examine the link between functional distinctiveness, extinction risk, and vulnerability to fishing.

## 2. Methods

### (a) Study area

The North Sea is a European epicontinental sea covering 750 000 km$^2$, connected to the Atlantic Ocean through the English Channel in the South and the Norwegian Sea in the North [35]. There is strong environmental and ecological heterogeneity between the northern and southern North Sea, particularly in terms of bathymetry and biodiversity [33]. The north is deeper, colder (most of the year) and has higher salinity, and the dominant fish species are Norway pout (*Trisopterus esmarkii*), haddock (*Melanogrammus aeglefinus*), saithe (*Pollachius virens*) and whiting (*Merlangius merlangus*) [36,37]. The south is characterized by warmer and shallower waters with lower salinity, higher primary productivity, and greater seasonal fluctuations in environmental conditions [38]. The dominant fish species are dab (*Limanda limanda*), plaice (*Pleuronectes platessa*), and herring (*Clupea harengus*) [36,37].

### (b) Fish survey and ecological traits

Fish abundance data were acquired through the International Bottom Trawling Survey (IBTS; https://datras.ices.dk/Data_products/Download/Download_Data_public.aspx), a fishery monitoring campaign conducted in the North Sea every year in January–February since 1983. For this monitoring, the North Sea is gridded in survey cells of 1° longitude by 0.5° latitude (as defined by the International Council for Exploitation of the Sea ICES), hereafter referred to as 'ICES rectangles'. Two 30-min hauls are performed in each ICES rectangle every year with a grand opening bottom trawl (GOV; 10-mm stretched mesh size in the codend) towed at an average speed of 4 knots. For each haul, all captured individuals are counted and identified at the most accurate taxonomic level possible. Over the entire North Sea and study period, the majority of taxa (116) were identified at species level and the remaining taxa (14) at genus level (hereafter, the term 'species' is used to refer to both levels of taxonomic identification). Species' abundances were standardized to numbers of individuals per km$^2$. The final fish abundance dataset included 130 species across 154 ICES rectangles over 33 years, thus providing a table with 130 columns and 4924 rows (some years did not include all 154 survey cells).

Fish ecological traits expected to be implicated in the response to environmental changes and/or fishing pressure were collected from public databases [39–41] (electronic supplementary material, table S1). Altogether, we considered five continuous traits (trophic level, age and size at sexual maturity, fecundity and offspring size) and three categorical traits (water-column position, diet, and parental care) belonging to three main categories (habitat preference, trophic ecology and life-history) (electronic supplementary material, table S1).

### (c) Functional distinctiveness identification

Functional distinctiveness Di is an index quantifying how functionally dissimilar a given species is on average compared to all other species in the regional pool [6,42]:

$$Di = \frac{\sum_{j=1, j \neq i}^{N} d_{ij}}{N - 1}$$

with $N$ the total number of species in the North Sea pool, and $d_{ij}$ the dissimilarity between species $i$ and $j$. We used Gower distance as a dissimilarity measure because our analyses used both continuous and categorical traits [43,44]. Functional dissimilarity was standardized so that distinctiveness ranged between 0 and 1. A functionally distinct species, i.e. with a high value of $Di$, corresponds to a species with highly original trait values compared to the rest of the regional species pool. Spearman correlation tests were performed to assess the relationship between each continuous trait and species functional distinctiveness. For categorical traits, differences in functional distinctiveness between trait modalities were tested using Wilcoxon *post hoc* tests. In the present study, we focused on functionally common and functionally distinct species by selecting the first and last quartiles of distinctiveness, respectively. The first group (Q1) included the most functionally common species while the fourth group (Q4) included the most functionally distinct species.

### (d) Vulnerability of functionally distinct species

We obtained species' IUCN (International Union for Conservation of Nature) statuses [45] using the *rredlist* R package [46] and compared the statuses across functional distinctiveness groups. The IUCN Red List classifies species according to their risk of extinction, based on criteria such as the rate of decline in population size, geographical range, number of mature individuals, and fragmentation of species distribution [45]. We performed $\chi^2$-tests of

homogeneity to compare the proportion of IUCN statuses between distinctiveness groups.

Species vulnerability to fishing is defined as the inherent capacity to respond to fishing pressure based on species maximum growth rate and strength of density dependence [47]. To evaluate the vulnerability of each species to fishing, we used the vulnerability index developed by Cheung *et al.* [47] which ranges from 0 (least concern) to 100 (most vulnerable). These authors reviewed published literature describing known relationships between biological characteristics and vulnerability and developed a rule-based system for scoring species vulnerability. Their index takes into account maximum length, age at first maturity, the von Bertalanffy growth parameter, natural mortality rate, maximum age, geographical range, fecundity, and spatial behaviour strength. Published vulnerability scores for each species (calculated by Cheung *et al.*) were obtained from FishBase using the *rfishbase* R package.

The association between functional distinctiveness and vulnerability to fishing was tested using a Spearman rank correlation test. We also tested the associations between functional distinctiveness and IUCN status and vulnerability to fishing using phylogenetic regressions (*phylolm* R package [48]), which account for non-independence among phylogenetic lineages. These analyses were performed for a subset of 93 species currently available in the phylogeny from the Fish Tree of Life (*fishtree* R package [49]).

## (e) Environmental parameters and fishing pressure

The North Sea is subject to two major climatic cycles: the North Atlantic Oscillation (NAO) and the Atlantic Multidecadal Oscillation (AMO). The NAO is an alternation of air movement over the Arctic and Icelandic regions, inducing changes in sea-level pressure, wind, temperature and precipitation in the North Atlantic [50]. The AMO is related to cycles in sea surface temperature (SST) that last several decades [51]. These environmental drivers are known to affect biodiversity, particularly fish and plankton communities [52,53]. We extracted one value of AMO and one value of NAO per year for the whole North Sea from the website of the National Oceanic and Atmospheric Administration (NOAA, US; NAO values: www.cpc.ncep.noaa.gov/products/precip/CWlink/pna/nao_index.html; AMO values: https://www.psl.noaa.gov/data/timeseries/AMO/).

Local environmental parameters included depth, phytoplankton biomass, sea surface temperature, salinity and bed shear stress. Sea surface temperature SST (°C) was extracted from the HadISST1 database of the Hadley Centre for Climate Prediction and Research [54]. Surface salinity (SSS; PSU: Practical Salinity Unit) was obtained from the NORWECOM biogeochemical model (NORWegian ECOlogical Model) [55]. Bed shear stress data (Bstress; water velocity over the bottom) were obtained from a 3D hydrodynamic model [56]. Phytoplankton biomass was estimated with the Phytoplankton Colour Index (PCI), derived from a Continuous Plankton Recorder towed by ships. All PCI values were extracted from the Continuous Plankton Recorder Survey dataset (https://www.cprsurvey.org/).

We quantified fishing pressure based on the trawling effort (number of hours) per ICES rectangle each year from 1985 to 2015 [57]. Trawling effort data were obtained from a reconstructed database integrating seven time series of historical trawling effort across the entire North Sea [57]. The full list of data sources is provided in electronic supplementary material, appendix 1.

## (f) Spatio-temporal dynamics of functional distinctiveness

We first mapped the spatial distribution of functional distinctiveness, in terms of species richness and total abundance of the species belonging to the functional distinctiveness quartiles Q1 (most common) and Q4 (most distinct). The abundances of all species in each group were summed for each ICES rectangle after calculating an average abundance per species per ICES rectangle for the entire time series. While we did not directly compare Q1 and Q4 species groups in this study, the abundances of functionally distinct species (Q4) were similar to the abundances of functionally common species (Q1) (electronic supplementary material, figure S1).

We designed three complementary redundancy analyses (RDA) [43] to analyse how the spatial and temporal dynamics of abundances in functional distinctiveness groups related to environmental variables and fishing pressure. The first RDA examined spatial dynamics across ICES rectangles, the second examined temporal dynamics for the overall North Sea ecosystem, and the third examined whether temporal changes varied across space. For the first RDA, we used the mean total abundance of Q1 and Q4 groups and mean values of environmental variables and fishing pressure per ICES rectangle (over the entire study period) (question 1 in electronic supplementary material, table S2). For the second RDA, we used the annual total abundance of Q1 and Q4 groups for the entire North Sea and mean annual environmental variables and fishing pressure (question 2 in electronic supplementary material, table S2). For the third RDA, temporal trends of the most common (Q1) and most distinct (Q4) species (quantified by Spearman correlation coefficients of abundance versus time in each ICES rectangle) were related to spatial differences (time-averaged) in environmental variables and fishing pressure (question 3 in electronic supplementary material, table S2).

All statistical analyses were performed with the R software (version 3.5.2; R Development Core Team, 2018) and all abundance data were log-transformed prior to analyses.

# 3. Results

## (a) Characterization of functionally distinct species

In the North Sea, the five most functionally distinct species were, in descending order, the tope shark ($Di = 0.60$; *Galeorhinus galeus*), spiny dogfish ($Di = 0.55$; *Squalus acanthias*), smooth-hound ($Di = 0.50$; *Mustelus* spp.), conger eel ($Di = 0.45$; *Conger conger*) and large castagnole ($Di = 0.45$; *Brama brama*) (electronic supplementary material, table S3). The five most functionally common species were, in increasing order of distinctiveness, the tub gurnard ($Di = 0.21$; *Chelidonichthys lucerna*), the grey gurnard ($Di = 0.21$; *Eutrigla gurnardus*), the fourbeard rockling ($Di = 0.21$; *Enchelyopus cimbrius*), the striped red mullet ($Di = 0.21$; *Mullus surmuletus*) and the scaldfish ($Di = 0.21$; *Arnoglossus* spp.) (electronic supplementary material, table S3).

Functionally distinct species had larger offspring, larger size at first maturity, higher age at first maturity, higher trophic level and lower fecundity than other species (electronic supplementary material, figure S2). Functionally distinct species were also primarily benthopelagic, pelagic or reef-associated, whereas demersal species (i.e. fishes living and feeding near the sea bottom) were more functionally common. Benthivorous species were generally less distinct than those with other diets, as were species providing little or no parental care.

## (b) Vulnerability of functionally distinct species

We found no significant link between species functional distinctiveness and IUCN status ($\chi^2$-test; *p*-value = 0.682; electronic supplementary material, figure S3); however, the

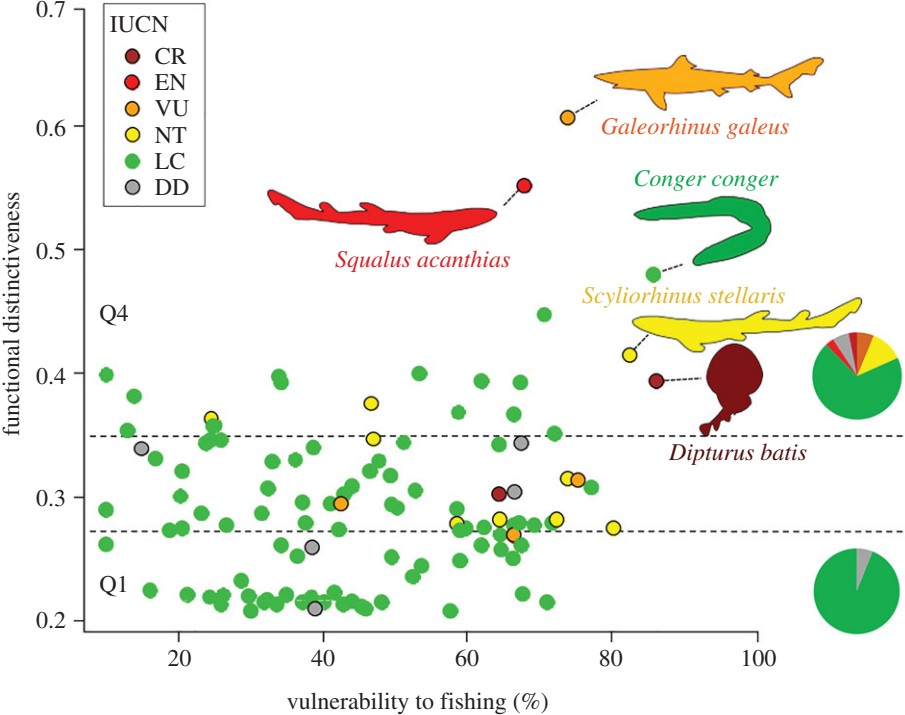

**Figure 1.** Relationship between species' vulnerability to fishing and functional distinctiveness. IUCN status is represented by colours (CR = critically endangered, dark red; EN = endangered, red; VU = vulnerable, orange; NT = near-threatened, yellow; LC = low concern, green; DD = data deficient, grey). The horizontal dotted lines indicate the delineation of the two groups of functional distinctiveness (Q1 = Quartile 1, functionally common species; Q4 = Quartile 4, functionally distinct species). The two pie charts show the proportion of species belonging to each IUCN category in the two groups of functional distinctiveness. (Online version in colour.)

proportion of threatened and endangered species (i.e. CR, EN and VU IUCN categories) increased from 0% for functionally common species to nearly 25% for functionally distinct species (figure 1).

We did not find any significant relationship between species' functional distinctiveness and vulnerability to fishing for the entire North Sea species pool (Spearman correlation test: $\rho = 0.14$, $p = 0.15$; figure 1). However, a few species had particularly high distinctiveness and high vulnerability in comparison to the entire species pool (figure 1), such as the tope shark (*Galeorhinus galeus*), spiny dogfish (*Squalus acanthias*) and common skate (*Dipturus batis*), which not only are highly vulnerable to fishing, but are also classified as vulnerable (VU), endangered (EN) and critically endangered (CR), respectively.

We also tested the associations between functional distinctiveness and IUCN status and vulnerability to fishing using phylogenetic regressions which account for non-independence among phylogenetic lineages. These analyses revealed no significant relationship between functional distinctiveness and IUCN status (electronic supplementary material, figure S3) or between functional distinctiveness and vulnerability to fishing (figure 1).

### (c) Spatio-temporal dynamics of functionally distinct species

The species richness and total abundance of functionally distinct species were higher in deeper, colder, and more saline environments (i.e. northern North Sea), while functionally common species were found in shallower, warmer environments with higher phytoplankton biomass, trawling effort and bed shear stress (i.e. southern North Sea; figure 2

and electronic supplementary material, figure S4). However, it should be noted that the latitudinal gradient for functionally distinct species (Q4) was less pronounced than for functionally common species (Q1). In the spatial RDA (question 1 in electronic supplementary material, table S2), 52% of variance was explained by environmental variables and fishing pressure, with depth, sea surface temperature (SST), Phytoplankton Colour Index (PCI) and trawling effort best explaining the spatial distribution of functional distinctiveness group abundance (figure 3*a*).

At the scale of the entire North Sea, the total abundance of both functionally distinct and functionally common species increased over the last three decades (functionally distinct— Spearman correlation test: $\rho = 0.55$, $p < 0.001$; functionally common—Spearman correlation test: $\rho = 0.89$, $p < 0.001$; figure 2). Among the 33 functionally distinct species, 24 (i.e. 73%) have increased in abundance and thus contributed to the rebound. However, we must note that the remaining 27% (nine species) have declined, among which two are listed as threatened by the IUCN (common skate: *Dipturus batis*, CR; spiny dogfish: *Squalus acanthias*, EN). Environmental variables and fishing pressure explained 56% of variance in the temporal RDA (question 2 in electronic supplementary material, table S2), with a major contribution from increasing SST (figure 3*b*).

Environmental variables and fishing pressure explained about 49% of variance in the spatio-temporal RDA (question 3 in electronic supplementary material, table S2), with depth, PCI, SST and bed shear stress (Bstress) best explaining the spatio-temporal patterns of the two functional distinctiveness groups (figure 3*c*). The abundance of functionally distinct species thus increased most in environments characterized by warm, shallow waters, low salinity, high phytoplankton

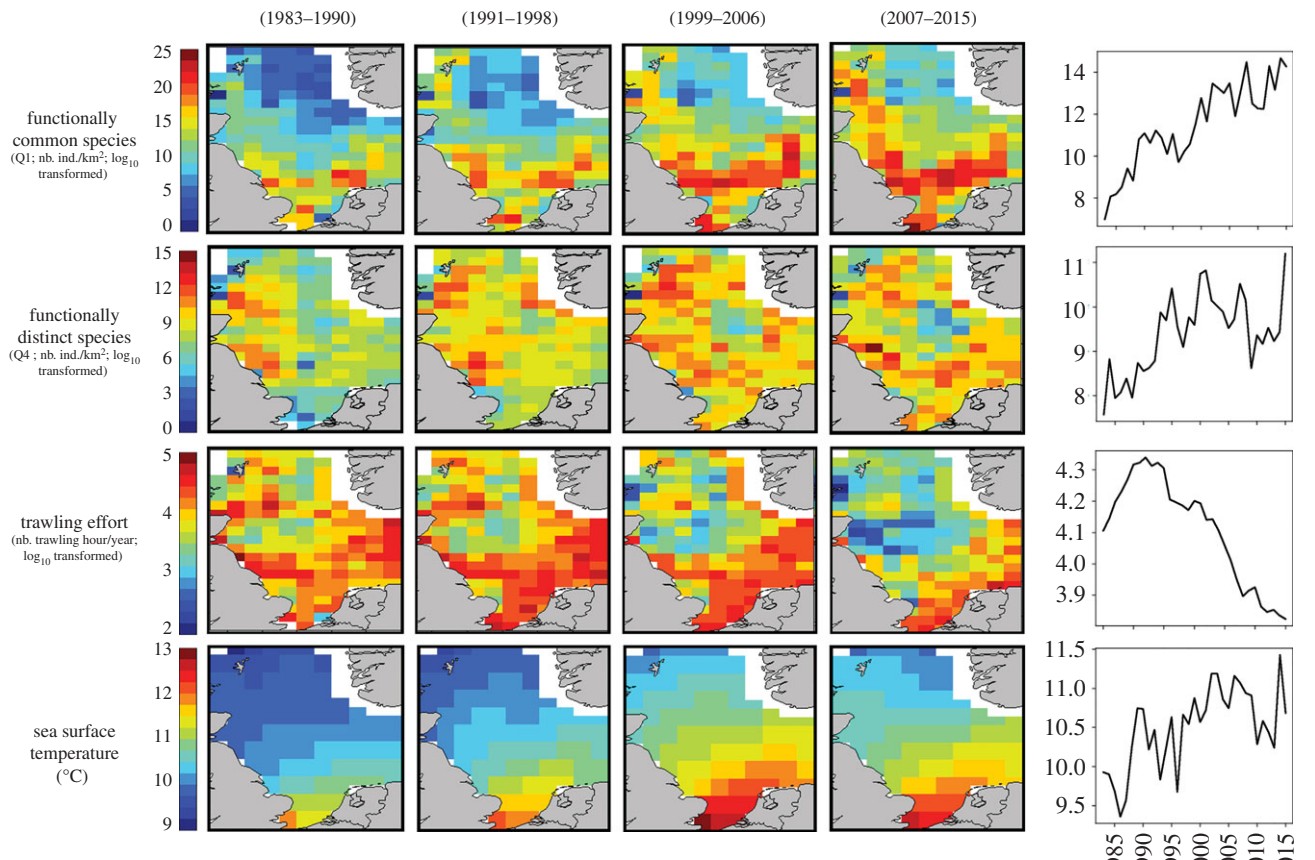

**Figure 2.** Temporal trends in the abundance of functionally common and functionally distinct species (log-transformed number of individuals per km$^2$), trawling effort (log-transformed trawling hours per year) and sea surface temperature (SST; in °C). The overall trends over the study period (1983–2015) are shown on the right panels. The maps present the average values of the variables calculated on sub-periods (1983–1990; 1991–1998; 1999–2006, 2007–2015) for the 154 ICES rectangles. (Online version in colour.)

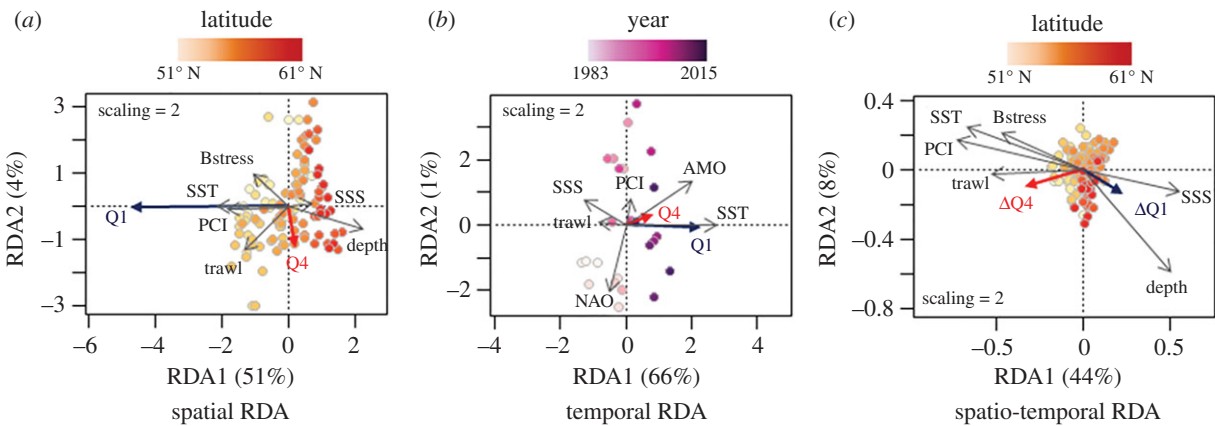

**Figure 3.** Biplots of redundancy analyses (RDA) conducted between fish abundances, environmental parameters and fishing pressure. Fish were separated into functional distinctiveness groups with Q1 representing the functionally common species and Q4 the functionally distinct species. (a) Spatial redundancy analysis between the mean total abundance of Q1 and Q4 groups per ICES rectangle and mean spatial environmental variables and fishing pressure. (b) Temporal redundancy analysis between the annual total abundance of Q1 and Q4 groups and mean annual environmental variables and fishing pressure. (c) Spatio-temporal redundancy analysis between the temporal trends (Spearman coefficients) of the total abundance of Q1 and Q4 groups per ICES rectangle and mean spatial environmental variables and fishing pressure. Acronyms: salinity 'SSS', shear stress 'Bstress', phytoplankton biomass 'PCI', sea surface temperature 'SST', North Atlantic Oscillation 'NAO' and Atlantic Multidecadal Oscillation 'AMO'. (Online version in colour.)

biomass and historically high trawling effort (i.e. the southern North Sea). Moreover, during the last three decades, the abundance of functionally distinct species has increased in 70% of the ICES rectangles where fishing has declined (figure 2). In contrast, the abundance of functionally common species increased the most in the northern North Sea (figure 2).

## 4. Discussion

This study reveals that fish functional distinctiveness in the North Sea is (i) markedly linked to certain life-history traits, (ii) geographically structured and (iii) strongly influenced by environmental conditions and fishing pressure. These results are of particular interest since functional

distinctiveness is increasingly recognized as a neglected facet of biodiversity that can be crucial for ecosystem stability and resilience [13]. This work highlights a positive message through the 'comeback' of functionally distinct species after decades of overexploitation until the 1990s fishing reduction. We found that functional distinctiveness was shaped by particular demographic strategies as well as habitat preferences. Pelagic species and benthopelagic species were more functionally distinct than demersal species. Functionally distinct species yield a few large eggs or offspring, reach sexual maturity later, and provide high parental care to their progeny, i.e. *K*-selected species [58]. *K*-selected species, with greater parental investment and offspring survivorship, are competitively superior in stable environments, and are often more resistant to short-term environmental changes, providing them an advantage for population growth [58,59]. However, in the long term, *K*-selected species are likely more vulnerable to environmental changes due to their long generation times, slow population growth and therefore lower adaptive capacity and resilience [31,58,59]. In the context of ongoing and future environmental changes, this raises a conservation concern since most functionally distinct species (i.e. species supporting unique functions in the ecosystem) in the North Sea are *K*-selected. Moreover, there is less functional redundancy in these species, which is a source of vulnerability to environmental changes. By contrast, most functionally common species in the North Sea are *r*-selected, which implies that they respond quickly to short-term environmental fluctuations, and have greater long-term adaptive capacity and resilience given their rapid population turnover and opportunistic strategies [31,58,59]. Our results then indicate that functionally distinct species are more vulnerable to climate change, and that marine ecosystems may homogenize toward functionally common, opportunistic species.

Although less pronounced for functionally distinct species, we found an opposite latitudinal gradient in the distribution of functionally distinct versus functionally common species in the North Sea. Functionally distinct species were especially found in the north (i.e. highest Q4 species richness; electronic supplementary material, figure S4) while functionally common species, with a clearer latitudinal gradient, were especially located in the southern North Sea (i.e. higher Q1 species richness and abundance; electronic supplementary material, figure S4). These results are consistent with previous studies showing spatial differences between fish communities in the southern and northern North Sea for both species and traits (e.g. [37]). Here we also detected a spatial segregation of fish taxa according to their functional distinctiveness, associated with a north–south contrast in environmental conditions and fishing pressure. Fishing effort was historically more intense in the southern North Sea [60], and fishing pressure on large species with late sexual maturity increased substantially between 1925 and 1996 [61]. As a consequence, fishing may have impacted *K*-selected species in the southern North Sea, thus potentially concentrating them in the north. We can therefore hypothesize that historical fishing has contributed to the current distribution of functional distinctiveness in the North Sea. Beyond fishing, environmental conditions have also shaped the distribution of functional distinctiveness, with strong patterns related to depth, temperature, and primary production. The southern North Sea also has greater environmental variability with higher seasonality

than the deeper waters of the northern North Sea, which could further explain the spatial distinction of *r*- and *K*-selected species, and thereby the functional distinctiveness patterns with *r*-selected (i.e. functionally common species) in the south, and *K*-selected (i.e. functionally distinct species) in the North. For instance, *K*-selected species are competitively superior in stable environments where populations reach carrying capacity and are limited by density dependence, whereas *r*-selected species flourish in dynamic and highly variable environments where density dependence is not limiting and rapid population growth provides a competitive advantage [62].

Both functionally distinct and functionally common species have increased in richness and abundance in the North Sea in parallel to progressive sea surface warming and a gradual decrease in fishing pressure since the 2000s. Interestingly, the temporal changes varied spatially, as functionally distinct species increased in abundance everywhere in the North Sea but primarily in the south where they were initially less abundant, while functionally common species increased in abundance primarily in the north. We can hypothesize that reduced fishing pressure on *K*-selected species allowed them to progressively rebuild their populations (especially in the southern North Sea), while *r*-selected species likely increased in the north, through northward range expansions and immigration from the North Atlantic [37]. Thus, our results highlight that both functionally common and distinct species have increased in the North Sea in response to rising temperatures and decreased fishing since the mid-1990s; however, decreased fishing pressure appears particularly important to the rebound in functionally distinct species. Because of their putative contribution to ecosystem stability, this rebound of functionally distinct species is probably one of the most positive consequences of the Common Fisheries Policy applied since the end of the 1970s. The reduction in catch quotas combined with improved gear selectivity has most likely favoured the rebound of functionally distinct species, which are known to be particularly sensitive to the type of trawling gear [63]. Although this study reveals one of the positive consequences of fishing reduction on biodiversity, efforts in ecosystem approach to fisheries management (EAFM) must be maintained as some of the most functionally distinct species, including threatened and vulnerable species such as common skate and spiny dogfish, have declined in abundance over the last 30 years.

While *K*-selected species are expected to be vulnerable to long-term, gradual climate change, they may have high resilience to initial or short-term environmental changes [59,64]. Thus, while ongoing and future warming could ultimately reduce the abundance of functionally distinct species, it appears they have been resilient to temperature rises over the last three decades. Functionally distinct species may have also benefited from warming through increased metabolic and reproductive rates as well as warming-induced trophic cascades [65]. For instance, the substantial increase in small, rapidly growing species, and general increase in overall fish abundance, may have increased prey availability for functionally distinct species, many of which are large-bodied and have high trophic levels.

Many of the most functionally distinct species are highly vulnerable to fishing and some, such as the tope shark, spiny dogfish, and common skate, are considered threatened with extinction. These species have nearly disappeared from

several regions like the southern North Sea in the 1980s due to long-term overfishing [66,67]. In particular, the common skate is the largest skate in the world and was once highly abundant throughout the northeast Atlantic Ocean, but has been extirpated from much of its native range [68]. More generally, a decline in the abundance of all shark species occurred in the North Sea between 1970 and 1985 [69]. Among them, the spiny dogfish suffered major regional declines [70], estimated at 95% in the North-East Atlantic stock since 2000 [71]. The decline or disappearance of such species, which are both vulnerable and functionally distinct, could have catastrophic consequences on ecosystem functioning since, by definition, few if any other species have similar trait combinations, and functionally distinct species may support irreplaceable ecosystem roles [6,11,72,73]. For instance, the loss of functionally distinct species belonging to higher trophic levels (e.g. sharks and skates), could have important cascading effects on overall ecosystem structure through top-down control [74,75]. The loss of predators from aquatic ecosystems has been associated with food web reorganization, invasive species outbreaks, altered nutrient and carbon cycling, and even habitat modification [76], and the loss of such species in the North Sea could pose unforeseen risks to ecosystem functioning. Here, exploring functional distinctiveness uncovered new and important aspects of North Sea fish community dynamics, which should help in designing adapted conservation and management practices in the future. Our results suggest that long-term historical fishing pressure led to strong spatial structuring of functionally distinctiveness—still visible today—through trait-based targeting of fishing activities (i.e. larger, long-lived species [32]). Our results also suggest that the spatial distribution of functional distinctiveness has homogenized over time, in response to both sea surface warming and the progressive decrease of fishing pressure during the last three decades. Despite these important insights about functional distinctiveness in the North Sea, our results are limited to the set of traits and species examined. We must indeed remind here that all the conclusions depend on the initial choice of ecological traits [77], and that the functional distinctiveness of each species depends on the pool of species considered. For this reason, the distinctiveness values calculated here are specific to the fish species pool of the North Sea, as the most functionally distinct species in the North Sea may be common elsewhere and vice versa.

Classical indices of functional diversity have already been used for conservation purposes, such as for defining priorities between species or ecosystems [78,79]. Future studies in this area would benefit from considering functional distinctiveness [80], and not only focusing on the most visible aspects of trait diversity. Since using ecological traits may help assess the impact of species loss on ecosystem functioning, conservation measures should integrate species' traits, including functional distinctiveness, rather than species identity alone. In particular, the IUCN Red List is currently being used as a potent tool to guide conservation strategies. However, the establishment of such strategies does not directly consider species ecological traits and roles in ecosystem functioning. Including a functional component in management plans would likely provide more relevant species conservation statuses with respect to ecosystem functioning and equilibrium [81].

Ethics. All abundance data used in that study are collected and provided by the International Council for the Exploration of the Sea (ICES), which is a world leading marine science organization that aims to advance and share scientific understanding of marine ecosystems and the services they provide and to use this knowledge to generate state-of-the-art advice for meeting conservation, management, and sustainability goals. All trawling operations performed during the IBTS survey were authorized by local authorities in the various countries around the North Sea.

Data accessibility. All data and R code used in this study are freely available at: https://figshare.com/articles/figure/Murgier_et_al_R_code_and_data/12609425

Authors' contributions. J.M., M.M., A.M. and A.A. wrote the manuscript and performed data analysis, D.M., N.L., F.M. and C.V. critically revised the manuscript. A.A. coordinated the study. All authors gave final approval for publication and agree to be held accountable for the work performed therein.

Competing interests. This work is not associated with any competing interests.

Funding. This research is supported by the Fondation pour la Recherche sur la Biodiversité (FRB) and Electricité de France (EDF) in the context of the CESAB project 'Causes and consequences of functional rarity from local to global scales' (FREE). C.V. was supported by the European Research Council (ERC) Starting Grant Project 'ecophysiological and biophysical constraints on domestication in crop plants' (grant ERC-StG-2014-639706-CONSTRAINTS).

Acknowledgements. This research is supported by the Fondation pour la Recherche sur la Biodiversité (FRB) and Electricité de France (EDF) in the context of the CESAB project 'Causes and consequences of functional rarity from local to global scales' (FREE). We especially would like to acknowledge all persons involved in the North Sea IBTS survey. We thank the reviewers who improved the quality of this paper.

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
