## [Reviewer comments · Proceedings of the Royal Society B: Biological Sciences]

Review History

RSPB-2020-1600.R0 (Original submission)

Review form: Reviewer 1

Recommendation

Major revision is needed (please make suggestions in comments)

Scientific importance: Is the manuscript an original and important contribution to its field?

Good

General interest: Is the paper of sufficient general interest?

Good

Quality of the paper: Is the overall quality of the paper suitable?

Good

Is the length of the paper justified?

Yes

Should the paper be seen by a specialist statistical reviewer?

No

Do you have any concerns about statistical analyses in this paper? If so, please specify them explicitly in your report.

No

It is a condition of publication that authors make their supporting data, code and materials available - either as supplementary material or hosted in an external repository. Please rate, if applicable, the supporting data on the following criteria.

Is it accessible?

Yes

Is it clear?

Yes

Is it adequate?

Yes

Do you have any ethical concerns with this paper?

No

Comments to the Author

Dear authors, Your manuscript deals with an important conservation question which is related to how species with distinct functional trait values respond to environmental changes. I find your study well-designed and well written. I have no doubt that your findings will be of interest of this journal audience as they shed light in how species traits can help us to understand ecological processes and, ultimately, how we can use this to improve conservation practices. I have just one comment:

There is evidence that rare fish species held also more unique trait combinations (Leitão et al. 2016 Rare species contribute disproportionately to the functional structure of species assemblages Proceedings of the Royal Society B: Biological Sciences 283, 20160084.). In fact, you cited a number of studies that also show this. However, you did not mention nor discuss this in your paper. My point here is that "if functionally unique species are also rare" when you discriminate your species in two categories (1st and 4th quartile; respectively, less unique and more unique species) you should demonstrate that these are not also the more abundant and rare species, respectively. If they are, then, you have a sort of hidden effect which need, at least, be addressed and properly discussed. Considering this is important because, one may say that the effect size of increasing in abundance would be expected to be higher for rare species (compared to more abundant species) despite of their traits.

Thus, are functionally unique increasing "more than expected" given their lower abundance? If they are, this could be an evidence that such unique trait combinations allow them to respond faster or better to environmental changes. Or in the North Sea functionally unique species are not also the rare ones?

Minor comments:

line 18: "Functionally distinct species (i.e., species with trait combinations unlike others in the community)..." It would be clear to state: "Functionally distinct species (i.e., species with unique trait combinations in the community)..."

line 31: I would start a new phrase "Alarmingly, these species have not rebounded."

Figure S1: It is not correct to include a line to represent any correlation analysis. This gives the idea that you did a linear model, which was not the case, specially using a non-parametric index.

It would be best to remove the lines and present the correlation index in the upper right corner of each figure.

Review form: Reviewer 2 (Xingli Giam)

Recommendation

Major revision is needed (please make suggestions in comments)

Scientific importance: Is the manuscript an original and important contribution to its field?

Excellent

General interest: Is the paper of sufficient general interest?

Excellent

Quality of the paper: Is the overall quality of the paper suitable?

Good

Is the length of the paper justified?

Yes

Should the paper be seen by a specialist statistical reviewer?

No

Do you have any concerns about statistical analyses in this paper? If so, please specify them explicitly in your report.

Yes

It is a condition of publication that authors make their supporting data, code and materials available - either as supplementary material or hosted in an external repository. Please rate, if applicable, the supporting data on the following criteria.

Is it accessible?

Yes

Is it clear?

Yes

Is it adequate?

Yes

Do you have any ethical concerns with this paper?

No

Comments to the Author

Review of RSPB-2020-1600

In this manuscript, Murgier et al. examined the spatiotemporal dynamics in fish functional distinctiveness over the past ~30 years in the North Sea and aimed to link these dynamics to fishing pressure, climate warming, and environmental conditions. The manuscript addresses an interesting ecological concept (functional distinctiveness) and how large-scale and relevant anthropic stresses (fish, climate warming) affect functional distinctiveness. I found the manuscript to be concise and well-written and would be of interest to the ecological research

community and the general public.

I have a few methodological recommendations that can hopefully improve the paper. First, I suggest the authors use phylogenetic comparative methods (phylogenetic lm) to test the associations between (i) functional distinctiveness and IUCN categories; and (ii) functional distinctiveness and vulnerability to fishing. This can be easily implemented using the `phylolm` package in R and using the phylogeny available in the recently published Fish Tree of Life (`fishree` package in R). Accounting for phylogeny will help ensure that the standard errors are not underestimated and that the statistical inferences are correct given that residual variation will likely be phylogenetically related as traits are.

Second, I think the authors should reconsider the analysis examining whether reduction in fishing pressure and warming are associated with the rebound in functional distinctiveness (particularly in the southern North Sea). The spatiotemporal RDA (Fig 3c) suggests that fishing pressure and Q4 abundance-time Spearman correlation is positively associated, and not negatively associated, indicating the opposite of what the authors concluded. Further, by averaging fishing and SST over the entire temporal period for each cell as predictor variables, I do not think the analysis was able to truly examine the question of whether there is a rebound in functional distinctiveness following warming and reduced fishing. A more appropriate analysis would be one that relates the abundance temporal trend (slope would be a better metric of trend than the Spearman correlation coefficient) to temporal trends in warming and fishing pressure – a meta-analytic approach can take into account the uncertainty around the slope. Also it is unclear whether the abundances of all four distinctiveness quartiles (vs. only Q1 and Q4) were used as response variables in the RDA (see specific comments below) because only Q1 and Q4 vectors were shown in the RDA. If only Q1 and Q4 were used as response variables, I wonder if it would be more straightforward to run two sets of phylogenetic glms (one with Q1 as response and the other with Q4 as response).

Last, there needs to be more discussion on how warming (after examining the association between temporal trend in distinctiveness vs warming trend and trend in fishing pressure) may be affecting distinctiveness. Presently, there is a good discussion on the effect of fishing pressure but the discussion on warming effects is currently lacking.

I hope my comments are useful. I am signing my review for transparency. Please contact me if you have any questions about my methodological recommendations.

- Xingli Giam

Specific comments

L95. Can the authors provide a more specific URL? The present URL links to a general landing page and does not point to where the IBTS data were downloaded.

L111. I suggest adding a brief description of the analysis performed to examine differences in mean distinctiveness across ecological traits [e.g., different habitat affinities, trophic groups, and parental care categories (the analyses associated with Fig. S1)].

L120. I suggest changing to “We used Gower distance as a dissimilarity measure because our analyses used both continuous and categorical traits.”

L133. It is unclear what is the difference between the first analysis “...compare the proportion of IUCN statuses within each functional distinctiveness group” and the second analysis “...compare the proportion of IUCN status between distinctiveness groups”. In the results section 3.2, I understand the analysis as examining how distinctiveness values differed among different IUCN groups. A more straightforward and appropriate analysis will be to use

L138. I suggest the authors expand on the description of the species vulnerability metric. Please state the source of the data (L131-143) used to calculate the metric.

L155. The link to the AMO data is broken. Please provide a new updated link to the data.

L163. Should this be updated to be named the “Continuous Plankton Recorder Survey”? And perhaps, this link is better: <https://www.cprssurvey.org/>

L177. It is unclear whether the RDAs used all four quartiles of distinctiveness or only Q1 and Q4 as suggested by Fig. 3 and the results section at L235-236. Please clarify. I am guessing that it is the latter (only Q1 and Q4 as suggested in L123-124). If so, I suggest the authors clarify this by revising the language in Table S2: e.g., by changing “for the species belonging to each quartile of functional distinctiveness” to “for the species belonging to the first (Q1) and last (Q4) quartile of functional distinctiveness”. Also do revise sentences like the one on L182: e.g., change “each functional distinctiveness group” to “functional distinctiveness quartiles Q1 (most common) and Q4 (most distinct)” and others e.g., the one on L187.

L207. Insert a full stop after “spp”.

L210. Typo error. It should be “primarily” instead of “primary”.

L216. Provide the results of the chi-square test for homogeneity.

L229. Suggest inserting “more” before “functionally common species”.

Fig. 1. Figure 1 contains information regarding IUCN status, vulnerability to fishing, and functional distinctiveness, which are the three variables relevant to the analysis, which provides a good level of information to readers. Specifically, the figure plots functional distinctiveness against fishing vulnerability. I suggest a further improvement to the figure so that it provides information that is more directly relevant to the first analysis: the relationship between functional distinctiveness and IUCN status. To do that, I suggest adding an inset figure of boxplots of distinctiveness values in different IUCN categories (i.e., like the boxplots in Figure S1).

Table S3. Suggest changing “Reparation” to “List”.

L235. How about the trends in the Q2 and Q3 species? Did the abundance of Q2 and Q3 species also rebound?

Fig. 3. Interesting that the dropoff in the variance explained by RDA2 (2-8%) compared to RDA1 (44-67%) was so great. This has implications for the interpretation and inference: that it is more important to look at the trends and associations with respect to the x-axis (RDA1).

Fig. 3. I also suggest adding points (of each observation) to the RDA plots, i.e., make it a triplot. I understand that it may make the figure a little messier and busier, but including the observations [individual ICE rectangles in (a) and (c) with color gradient reflecting latitude; individual years in (b) with color gradient reflecting year] can help the reader interpret the gradients and associations with environmental variables and functional distinctiveness groups better.

Also, did the RDA plots use Type 1 or Type 2 scaling? To the best of my knowledge, Type 2 scaling is appropriate when the aim is to illustrate associations between the response variables (i.e. relationships between Q1 and Q4 abundances), while Type 1 scaling is appropriate when the aim is to illustrate distances or similarities among observations. One option is to present Type 1 scaling in the full text, since the authors want to discuss the lat and time gradient; and present Type 2 scaling in the SI, where the focus can be on the correlation between Q1 and Q4 responses.

L233. According to Fig. 3a, PCI is also important as it has a highly negative RDA1 value (only slightly different from SST).

L261. “Functional distinctiveness was highly related to particular demographic strategies as well as habitat preferences” – This point is a given because functional distinctiveness is based on these traits and habitat affinities so distinctiveness has to be based on them. I rephrasing the sentence by replacing “was highly related to” with “was contributed by”.

L276. Suggest replacing “support that” with “indicate that”.

L282. I don’t think the authors need to cite the figure because this is a result that has already been presented in the results section.

L228-231 and L279-282. Suggest tightening up the sentence here. What does more frequent mean? Importantly, looking at Fig. S2, there is not a clear gradient in the species richness of functionally distinct species from north to south (perhaps a very slight gradient; Fig. S2b), and for dominance (Fig. S2d), the pixels of highest dominance seem to be in middle latitudes between the north and south. Same pattern is observed with abundance and numerical dominance (Fig. S2f and S2h respectively). I think the authors need to acknowledge that the N-S gradient for Q4 species isn’t as strong as the gradient for functionally common species.

L297-282. I think the manuscript is missing a discussion on the mechanisms by which climate warming may have affected the spatiotemporal dynamics of functional distinctiveness. The emphasis was on fishing pressure here, but I think readers would be interested to read about how warming may also be important, given that warming was in the title of the paper.

L288-296. I think the two hypotheses here need to be supported by previous studies, if possible. Also, the paper will benefit from a brief discussion on the potential mechanisms behind the association between r-K gradient, and gradient of seasonality/deeper waters (environmental stability).

L297-307. I am not sure if the data and analyses support the idea that reduced fishing is particularly important for the rebound in functionally distinct species particularly in the southern North Sea. Looking at the second row of panels in Fig. 2, I think the abundance of Q4 species has increased in both northern and southern regions. It is not particularly apparent that the rebound in the south is more pronounced than that in the north, perhaps by a little bit. Also in Fig. 3, Q4 is positively related to “trawl”, which suggests that sites with higher “trawl” values (i.e., higher fishing pressure), are also the sites where the abundance of functionally distinct species was increasing with time.

L363. Typo error. It should be “Funding” rather than “Founding”.

Decision letter (RSPB-2020-1600.R0)

06-Oct-2020

Dear Dr Auber:

Your manuscript has now been peer reviewed and the reviews have been assessed by an Associate Editor. The reviewers’ comments (not including confidential comments to the Editor) and the comments from the Associate Editor are included at the end of this email for your reference. As you will see, the reviewers and the Editors have raised some concerns with your manuscript and we would like to invite you to revise your manuscript to address them.

Research ethics:

Use of animals and field studies:

It is a condition of publication that you make available the data and research materials supporting the results in the article. Please see our Data Sharing Policies (<https://royalsociety.org/journals/authors/author-guidelines/#data>). Datasets should be deposited in an appropriate publicly available repository and details of the associated accession number, link or DOI to the datasets must be included in the Data Accessibility section of the article (<https://royalsociety.org/journals/ethics-policies/data-sharing-mining/>). Reference(s) to datasets should also be included in the reference list of the article with DOIs (where available).

Please submit a copy of your revised paper within three weeks. If we do not hear from you within this time your manuscript will be rejected. If you are unable to meet this deadline please let us know as soon as possible, as we may be able to grant a short extension.

Best wishes,
Dr Daniel Costa
mailto:proceedingsb@royalsociety.org

Associate Editor
Board Member: 1
Comments to Author:

Using yearly fish abundance data in the North Sea surveyed in the last three decades, together with data of species traits, fishing pressure and environmental changes, the authors identified functionally common and distinct species, and explored the spatiotemporal variations in species abundance in response to environmental and fishing pressure changes. They emphasized on the difference in the abundance changes of functionally common and distinct species. With these analyses, they addressed how functional distinctiveness of fish communities rebounded after the reduction in fishing pressure and also in response to environmental change. This manuscript has been well written, and the massive dataset used here is highly valuable for future studies. Two reviewers have reviewed this manuscript, and both of them acknowledged the merits of this study. I agree with them, and think that this manuscript has high potential to contribute to the field. However, both reviewers have provided very critical and useful comments. I would like to reinforce the comments of reviewer 2 on analytical methods. I found that the suggestions from the reviewer are very relevant. I suggest the authors to consider these comments in their revision.

Reviewer(s)' Comments to Author:
Referee: 1

Comments to the Author(s)

Dear authors, Your manuscript deals with an important conservation question which is related to how species with distinct functional trait values respond to environmental changes. I find your study well-designed and well written. I have no doubt that your findings will be of interest of this journal audience as they shed light in how species traits can help us to understand ecological processes and, ultimately, how we can use this to improve conservation practices. I have just one comment:

There is evidence that rare fish species held also more unique trait combinations (Leitão et al. 2016 Rare species contribute disproportionately to the functional structure of species assemblages *Proceedings of the Royal Society B: Biological Sciences* 283, 20160084.). In fact, you cited a number of studies that also show this. However, you did not mention nor discuss this in your paper. My point here is that "if functionally unique species are also rare" when you discriminate your species in two categories (1st and 4th quartile; respectively, less unique and more unique species) you should demonstrate that these are not also the more abundant and rare species, respectively. If they are, then, you have a sort of hidden effect which need, at least, be addressed and properly discussed. Considering this is important because, one may say that the effect size of increasing in abundance would be expected to be higher for rare species (compared to more abundant species) despite of their traits.

Thus, are functionally unique increasing "more than expected" given their lower abundance? If they are, this could be an evidence that such unique trait combinations allow them to respond faster or better to environmental changes. Or in the North Sea functionally unique species are not also the rare ones?

Minor comments:

line 18: "Functionally distinct species (i.e., species with trait combinations unlike others in the community)...". It would be clear to state: "Functionally distinct species (i.e., species with unique trait combinations in the community)...".

line 31: I would start a new phrase "Alarmingly, these species have not rebounded."

Figure S1: It is not correct to include a line to represent any correlation analysis. This gives the idea that you did a linear model, which was not the case, specially using a non-parametric index. It would be best to remove the lines and present the correlation index in the upper right corner of each figure.

Referee: 2

Comments to the Author(s)

Review of RSPB-2020-1600

In this manuscript, Murgier et al. examined the spatiotemporal dynamics in fish functional distinctiveness over the past ~30 years in the North Sea and aimed to link these dynamics to fishing pressure, climate warming, and environmental conditions. The manuscript addresses an interesting ecological concept (functional distinctiveness) and how large-scale and relevant anthropic stresses (fish, climate warming) affect functional distinctiveness. I found the manuscript to be concise and well-written and would be of interest to the ecological research community and the general public.

I have a few methodological recommendations that can hopefully improve the paper. First, I suggest the authors use phylogenetic comparative methods (phylogenetic lm) to test the associations between (i) functional distinctiveness and IUCN categories; and (ii) functional distinctiveness and vulnerability to fishing. This can be easily implemented using the *phylolm* package in R and using the phylogeny available in the recently published *Fish Tree of Life* (*fishtree* package in R). Accounting for phylogeny will help ensure that the standard errors are not underestimated and that the statistical inferences are correct given that residual variation will likely be phylogenetically related as traits are.

Second, I think the authors should reconsider the analysis examining whether reduction in fishing pressure and warming are associated with the rebound in functional distinctiveness (particularly in the southern North Sea). The spatiotemporal RDA (Fig 3c) suggests that fishing

pressure and Q4 abundance-time Spearman correlation is positively associated, and not negatively associated, indicating the opposite of what the authors concluded. Further, by averaging fishing and SST over the entire temporal period for each cell as predictor variables, I do not think the analysis was able to truly examine the question of whether there is a rebound in functional distinctiveness following warming and reduced fishing. A more appropriate analysis would be one that relates the abundance temporal trend (slope would be a better metric of trend than the Spearman correlation coefficient) to temporal trends in warming and fishing pressure – a meta-analytic approach can take into account the uncertainty around the slope. Also it is unclear whether the abundances of all four distinctiveness quartiles (vs. only Q1 and Q4) were used as response variables in the RDA (see specific comments below) because only Q1 and Q4 vectors were shown in the RDA. If only Q1 and Q4 were used as response variables, I wonder if it would be more straightforward to run two sets of phylogenetic GLMs (one with Q1 as response and the other with Q4 as response).

Last, there needs to be more discussion on how warming (after examining the association between temporal trend in distinctiveness vs warming trend and trend in fishing pressure) may be affecting distinctiveness. Presently, there is a good discussion on the effect of fishing pressure but the discussion on warming effects is currently lacking.

I hope my comments are useful. I am signing my review for transparency. Please contact me if you have any questions about my methodological recommendations.

- Xingli Giam

Specific comments

L95. Can the authors provide a more specific URL? The present URL links to a general landing page and does not point to where the IBTS data were downloaded.

L111. I suggest adding a brief description of the analysis performed to examine differences in mean distinctiveness across ecological traits [e.g., different habitat affinities, trophic groups, and parental care categories (the analyses associated with Fig. S1)].

L120. I suggest changing to “We used Gower distance as a dissimilarity measure because our analyses used both continuous and categorical traits.”

L133. It is unclear what is the difference between the first analysis “...compare the proportion of IUCN statuses within each functional distinctiveness group” and the second analysis “...compare the proportion of IUCN status between distinctiveness groups”. In the results section 3.2, I understand the analysis as examining how distinctiveness values differed among different IUCN groups. A more straightforward and appropriate analysis will be to use

L138. I suggest the authors expand on the description of the species vulnerability metric. Please state the source of the data (L131-143) used to calculate the metric.

L155. The link to the AMO data is broken. Please provide a new updated link to the data.

L163. Should this be updated to be named the “Continuous Plankton Recorder Survey”? And perhaps, this link is better: <https://www.cprsurvey.org/>

L177. It is unclear whether the RDAs used all four quartiles of distinctiveness or only Q1 and Q4 as suggested by Fig. 3 and the results section at L235-236. Please clarify. I am guessing that it is the latter (only Q1 and Q4 as suggested in L123-124). If so, I suggest the authors clarify this by revising the language in Table S2: e.g., by changing “for the species belonging to each quartile of functional distinctiveness” to “for the species belonging to the first (Q1) and last (Q4) quartile of functional distinctiveness”. Also do revise sentences like the one on L182: e.g., change “each

functional distinctiveness group” to “functional distinctiveness quartiles Q1 (most common) and Q4 (most distinct)” and others e.g., the one on L187.

L207. Insert a full stop after “spp”.

L210. Typo error. It should be “primarily” instead of “primary”.

L216. Provide the results of the chi-square test for homogeneity.

L229. Suggest inserting “more” before “functionally common species”.

Fig. 1. Figure 1 contains information regarding IUCN status, vulnerability to fishing, and functional distinctiveness, which are the three variables relevant to the analysis, which provides a good level of information to readers. Specifically, the figure plots functional distinctiveness against fishing vulnerability. I suggest a further improvement to the figure so that it provides information that is more directly relevant to the first analysis: the relationship between functional distinctiveness and IUCN status. To do that, I suggest adding an inset figure of boxplots of distinctiveness values in different IUCN categories (i.e., like the boxplots in Figure S1).

Table S3. Suggest changing “Reparation” to “List”.

L235. How about the trends in the Q2 and Q3 species? Did the abundance of Q2 and Q3 species also rebound?

Fig. 3. Interesting that the dropoff in the variance explained by RDA2 (2-8%) compared to RDA1 (44-67%) was so great. This has implications for the interpretation and inference: that it is more important to look at the trends and associations with respect to the x-axis (RDA1).

Fig. 3. I also suggest adding points (of each observation) to the RDA plots, i.e., make it a triplot. I understand that it may make the figure a little messier and busier, but including the observations [individual ICE rectangles in (a) and (c) with color gradient reflecting latitude; individual years in (b) with color gradient reflecting year] can help the reader interpret the gradients and associations with environmental variables and functional distinctiveness groups better. Also, did the RDA plots use Type 1 or Type 2 scaling? To the best of my knowledge, Type 2 scaling is appropriate when the aim is to illustrate associations between the response variables (i.e. relationships between Q1 and Q4 abundances), while Type 1 scaling is appropriate when the aim is to illustrate distances or similarities among observations. One option is to present Type 1 scaling in the full text, since the authors want to discuss the lat and time gradient; and present Type 2 scaling in the SI, where the focus can be on the correlation between Q1 and Q4 responses.

L233. According to Fig. 3a, PCI is also important as it has a highly negative RDA1 value (only slightly different from SST).

L261. “Functional distinctiveness was highly related to particular demographic strategies as well as habitat preferences” – This point is a given because functional distinctiveness is based on these traits and habitat affinities so distinctiveness has to be based on them. I rephrasing the sentence by replacing “was highly related to” with “was contributed by”.

L276. Suggest replacing “support that” with “indicate that”.

L282. I don’t think the authors need to cite the figure because this is a result that has already been presented in the results section.

L228-231 and L279-282. Suggest tightening up the sentence here. What does more frequent mean? Importantly, looking at Fig. S2, there is not a clear gradient in the species richness of functionally distinct species from north to south (perhaps a very slight gradient; Fig. S2b), and for dominance (Fig. S2d), the pixels of highest dominance seem to be in middle latitudes between the north and south. Same pattern is observed with abundance and numerical dominance (Fig. S2f and S2h).

respectively). I think the authors need to acknowledge that the N-S gradient for Q4 species isn't as strong as the gradient for functionally common species.

L297-282. I think the manuscript is missing a discussion on the mechanisms by which climate warming may have affected the spatiotemporal dynamics of functional distinctiveness. The emphasis was on fishing pressure here, but I think readers would be interested to read about how warming may also be important, given that warming was in the title of the paper.

L288-296. I think the two hypotheses here need to be supported by previous studies, if possible. Also, the paper will benefit from a brief discussion on the potential mechanisms behind the association between r-K gradient, and gradient of seasonality/deeper waters (environmental stability).

L297-307. I am not sure if the data and analyses support the idea that reduced fishing is particularly important for the rebound in functionally distinct species particularly in the southern North Sea. Looking at the second row of panels in Fig. 2, I think the abundance of Q4 species has increased in both northern and southern regions. It is not particularly apparent that the rebound in the south is more pronounced than that in the north, perhaps by a little bit. Also in Fig. 3, Q4 is positively related to "trawl", which suggests that sites with higher "trawl" values (i.e., higher fishing pressure), are also the sites where the abundance of functionally distinct species was increasing with time.

L363. Typo error. It should be "Funding" rather than "Founding".

Author's Response to Decision Letter for (RSPB-2020-1600.R0)

See Appendix A.

RSPB-2020-1600.R1 (Revision)

Review form: Reviewer 2 (Xingli Giam)

Recommendation

Accept with minor revision (please list in comments)

Scientific importance: Is the manuscript an original and important contribution to its field?

Excellent

General interest: Is the paper of sufficient general interest?

Excellent

Quality of the paper: Is the overall quality of the paper suitable?

Excellent

Is the length of the paper justified?

Yes

Should the paper be seen by a specialist statistical reviewer?

No

Do you have any concerns about statistical analyses in this paper? If so, please specify them explicitly in your report.

No

It is a condition of publication that authors make their supporting data, code and materials available - either as supplementary material or hosted in an external repository. Please rate, if applicable, the supporting data on the following criteria.

Is it accessible?

Yes

Is it clear?

Yes

Is it adequate?

Yes

Do you have any ethical concerns with this paper?

No

Comments to the Author

Review of RSPB-2020-1600.R1

The authors have addressed most of my previous concerns and suggestions. I think this is an important paper that should be published. However, I still have comments about the interpretation of spatiotemporal RDA and the discussion of spatiotemporal trends in functionally common vs. functionally rare species. Perhaps, it stems from my misunderstanding of the analyses, but I hope the authors can consider my comments and revise the manuscript accordingly if applicable.

L254-260. I suggest the authors include another sentence after the first sentence discussing variables that correlate with the increase of functionally common species. Perhaps it's enough to say that it's the opposite variables since the change in distinct vs. common species load on two opposite ends of spatiotemporal RDA axis 1.

The second sentence "Moreover, during the last three decades..." and the third sentence "In contrast, the abundance of functionally common species..." do not link up very well. I think here the authors want to first clarify that although the average trawling pressure is highest in the southern regions in the past 30 years, the trawling pressure there have decreased through time in this time period. This will give better context to the discussion in the Discussion section.

"Moreover, during the last three decades..." I think this statement needs a comparison. For example, in the same rectangles where fishing has declined, what is the percentage of these rectangles that have seen an increase in the abundance of functionally common species?

In the last sentence "In contrast, the abundance of functionally common species...", I actually do not agree with this finding. If you look at Fig. 2, first row, the figure referenced by this statement, I think the rectangles in the southern region increased in abundance (from yellow and orange to dark red) of functionally common species as much as the northern region (from blue to light blue green). I think this statement needs to be revised as well as the associated discussion (paragraph starting from L310, particularly L316; based on Fig. 2, I would say that functionally common, r-selected species also increased also in the south). I suggest comparing the value of the Spearman correlation averaged across all northern rectangles vs. the mean value across all southern rectangles to confirm spatial differences in temporal abundance change.

- Xingli Giam

Decision letter (RSPB-2020-1600.R1)

07-Dec-2020

Dear Dr Auber

I am pleased to inform you that your Review manuscript RSPB-2020-1600.R1 entitled "Rebound in functional distinctiveness following warming and reduced fishing in the North Sea" has been accepted for publication in Proceedings B.

The referee(s) do not recommend any further changes. Therefore, please proof-read your manuscript carefully and upload your final files for publication. Because the schedule for publication is very tight, it is a condition of publication that you submit the revised version of your manuscript within 7 days. If you do not think you will be able to meet this date please let me know immediately.

To upload your manuscript, log into <http://mc.manuscriptcentral.com/prsb> and enter your Author Centre, where you will find your manuscript title listed under "Manuscripts with Decisions." Under "Actions," click on "Create a Revision." Your manuscript number has been appended to denote a revision.

You will be unable to make your revisions on the originally submitted version of the manuscript. Instead, upload a new version through your Author Centre.

- 1) A text file of the manuscript (doc, txt, rtf or tex), including the references, tables (including captions) and figure captions. Please remove any tracked changes from the text before submission. PDF files are not an accepted format for the "Main Document".
- 2) A separate electronic file of each figure (tiff, EPS or print-quality PDF preferred). The format should be produced directly from original creation package, or original software format. Please note that PowerPoint files are not accepted.

- 3) Electronic supplementary material: this should be contained in a separate file from the main text and the file name should contain the author's name and journal name, e.g
authorname_procb_ESM_figures.pdf

All supplementary materials accompanying an accepted article will be treated as in their final form. They will be published alongside the paper on the journal website and posted on the online figshare repository. Files on figshare will be made available approximately one week before the accompanying article so that the supplementary material can be attributed a unique DOI. Please see: <https://royalsociety.org/journals/authors/author-guidelines/>

- 4) Data-Sharing and data citation

It is a condition of publication that data supporting your paper are made available. Data should be made available either in the electronic supplementary material or through an appropriate repository. Details of how to access data should be included in your paper. Please see <https://royalsociety.org/journals/ethics-policies/data-sharing-mining/> for more details.

<http://datadryad.org/submit?journalID=RSPB&manu=RSPB-2020-1600.R1> which will take you to your unique entry in the Dryad repository.

Once again, thank you for submitting your manuscript to Proceedings B and I look forward to receiving your final version. If you have any questions at all, please do not hesitate to get in touch.

Sincerely,
Dr Daniel Costa
Editor, Proceedings B
<mailto:proceedingsb@royalsociety.org>

Associate Editor Board Member: 1

Comments to Author:

The authors have made substantial revisions following the comments and suggestions of the two reviewers. I largely agree with the revisions. In general, I think this manuscript provides a very novel study about the long-term recovery of functional distinctiveness of fishes in the North Sea. The results of this manuscript shed provide new insight for more efficient conservation of fish biodiversity in the North Sea. One of the reviewers reviewed this manuscript again, and is satisfied with the revision. This reviewer has provided some more suggestions to improve the writing of the manuscript. I suggest that authors to take these comments into their revision.

Reviewer(s)' Comments to Author:

Referee: 2

Comments to the Author(s)
Review of RSPB-2020-1600.R1

The authors have addressed most of my previous concerns and suggestions. I think this is an important paper that should be published. However, I still have comments about the interpretation of spatiotemporal RDA and the discussion of spatiotemporal trends in functionally common vs. functionally rare species. Perhaps, it stems from my misunderstanding of the analyses, but I hope the authors can consider my comments and revise the manuscript accordingly if applicable.

L254-260. I suggest the authors include another sentence after the first sentence discussing variables that correlate with the increase of functionally common species. Perhaps it's enough to say that it's the opposite variables since the change in distinct vs. common species load on two opposite ends of spatiotemporal RDA axis 1.

The second sentence "Moreover, during the last three decades..." and the third sentence "In contrast, the abundance of functionally common species..." do not link up very well. I think here the authors want to first clarify that the although the average trawling pressure is highest in the southern regions in the past 30 years, the trawling pressure there have decreased through time in this time period. This will give better context to the discussion in the Discussion section.

"Moreover, during the last three decades..." I think this statement needs a comparison. For example, in the same rectangles where fishing has declined, what is the percentage of these rectangles that have seen an increase in the abundance of functionally common species?

In the last sentence "In contrast, the abundance of functionally common species...", I actually do not agree with this finding. If you look at Fig. 2, first row, the figure referenced by this statement,

I think the rectangles in the southern region increased in abundance (from yellow and orange to dark red) of functionally common species as much as the northern region (from blue to light blue green). I think this statement needs to be revised as well as the associated discussion (paragraph starting from L310, particularly L316; based on Fig. 2, I would say that functionally common, r-selected species also increased also in the south). I suggest comparing the value of the Spearman correlation averaged across all northern rectangles vs. the mean value across all southern rectangles to confirm spatial differences in temporal abundance change.

- Xingli Giam

Decision letter (RSPB-2020-1600.R2)

08-Dec-2020

Dear Dr Auber

I am pleased to inform you that your manuscript entitled "Rebound in functional distinctiveness following warming and reduced fishing in the North Sea" has been accepted for publication in Proceedings B.

Open Access

Paper charges

You are allowed to post any version of your manuscript on a personal website, repository or preprint server. However, the work remains under media embargo and you should not discuss it

with the press until the date of publication. Please visit <https://royalsociety.org/journals/ethics-policies/media-embargo> for more information.

Sincerely,
Editor, Proceedings B
<mailto:proceedingsb@royalsociety.org>

Appendix A

Referee: 1

Comments to the Author(s):

Dear authors,

Your manuscript deals with an important conservation question which is related to how species with distinct functional trait values respond to environmental changes. I find your study well-designed and well written. I have no doubt that your findings will be of interest of this journal audience as they shed light in how species traits can help us to understand ecological processes and, ultimately, how we can use this to improve conservation practices. I have just one comment: There is evidence that rare fish species held also more unique trait combinations (Leitão et al. 2016 Rare species contribute disproportionately to the functional structure of species assemblages Proceedings of the Royal Society B: Biological Sciences 283, 20160084.). In fact, you cited a number of studies that also show this. However, you did not mention nor discuss this in your paper. My point here is that "if functionally unique species are also rare" when you discriminate your species in two categories (1st and 4th quartile; respectively, less unique and more unique species) you should demonstrate that these are not also the more abundant and rare species, respectively. If they are, then, you have a sort of hidden effect which need, at least, be addressed and properly discussed. Considering this is important because, one may say that the effect size of increasing in abundance would be expected to be higher for rare species (compared to more abundant species) despite of their traits. Thus, are functionally unique increasing "more than expected" given their lower abundance? If they are, this could be an evidence that such unique trait combinations allow them to respond faster or better to environmental changes. Or in the North Sea functionally unique species are not also the rare ones? Thank you for that comment. A boxplot showing that common and distinct species (Q1 and Q4, respectively) have comparable abundance levels is now included in the supplementary material and a sentence is now added in the revised manuscript (l. 181-183 of the revised version): "While we did not directly compare Q1 and Q4 species groups in this study, the abundances of functionally distinct species (Q4) were similar to the abundances of functionally common species (Q1) (Figure S1).".

Minor comments:

line 18: "Functionally distinct species (i.e., species with trait combinations unlike others in the community)...". It would be clear to state: "Functionally distinct species (i.e., species with unique trait combinations in the community)...".

This has been changed as suggested.

line 31: I would start a new phrase "Alarmingly, these species have not rebounded."

Done accordingly.

Figure S1: It is not correct to include a line to represent any correlation analysis. This gives the idea that you did a linear model, which was not the case, specially using a non-parametric index. It would be best to remove the lines and present the correlation index in the upper right corner of each figure.

The regression lines have been removed from the figure and the correlation values have been added directly on the graphs.

Referee: 2

Comments to the Author(s) Review of RSPB-2020-1600

In this manuscript, Murgier et al. examined the spatiotemporal dynamics in fish functional distinctiveness over the past ~30 years in the North Sea and aimed to link these dynamics to fishing

pressure, climate warming, and environmental conditions. The manuscript addresses an interesting ecological concept (functional distinctiveness) and how large-scale and relevant anthropic stresses (fish, climate warming) affect functional distinctiveness. I found the manuscript to be concise and well-written and would be of interest to the ecological research community and the general public. I have a few methodological recommendations that can hopefully improve the paper. First, I suggest the authors use phylogenetic comparative methods (phylogenetic lm) to test the associations between (i) functional distinctiveness and IUCN categories; and (ii) functional distinctiveness and vulnerability to fishing. This can be easily implemented using the `phylolm` package in R and using the phylogeny available in the recently published Fish Tree of Life (`fishree` package in R). Accounting for phylogeny will help ensure that the standard errors are not underestimated and that the statistical inferences are correct given that residual variation will likely be phylogenetically related as traits are.

Thank you for this suggestion. We have now added phylogenetic regressions to test the association between functional distinctiveness and (i) IUCN status and (ii) vulnerability to fishing using the `phylolm` package and Fish Tree of Life phylogeny (L. 145-150 & 227-231). These analyses provided similar results as already presented, reinforcing that there was no significant association between distinctiveness and vulnerability to fishing (e.g., with the figure just blow) or IUCN status.

Second, I think the authors should reconsider the analysis examining whether reduction in fishing pressure and warming are associated with the rebound in functional distinctiveness (particularly in the southern North Sea). The spatiotemporal RDA (Fig 3c) suggests that fishing pressure and Q4 abundance-time Spearman correlation is positively associated, and not negatively associated, indicating the opposite of what the authors concluded. Further, by averaging fishing and SST over the entire temporal period for each cell as predictor variables, I do not think the analysis was able to truly examine the question of whether there is a rebound in functional distinctiveness following warming and reduced fishing. A more appropriate analysis would be one that relates the abundance temporal trend (slope would be a better metric of trend than the Spearman correlation coefficient) to temporal trends in warming and fishing pressure. A meta-analytic approach can take into account the uncertainty around the slope.

The aim of this spatiotemporal RDA is not to investigate the effect of an increase/decrease of explanatory variables on Q1 and Q4 (the temporal RDA has this objective). The spatiotemporal RDA aims to characterize in which 'average' environmental/fishing pressure conditions Q1 and Q4 species increased/decreased. The fact that fishing pressure and Q4 abundance-time Spearman correlation are positively associated indicates that Q4 species mostly increased in warmer rather than in colder environments (see l. 254-257 : "The abundance of functionally distinct species thus increased most in environments characterized by warm, shallow waters, low salinity, high phytoplankton biomass and historically high trawling effort (i.e., the southern North Sea)"). The effect of pressure variables trends on temporal trends of Q1 and Q4 are given by the second RDA (temporal RDA) (i.e., Fig 3b).

For better clarity we reworded the objective of the spatiotemporal RDA in table S2 as follows: “Are temporal trends in functional distinctiveness related to spatial environmental conditions?”.

Also it is unclear whether the abundances of all four distinctiveness quartiles (vs. only Q1 and Q4) were used as response variables in the RDA (see specific comments below) because only Q1 and Q4 vectors were shown in the RDA.

Thanks again for that important comment, and for providing suggestions. Yes, this is only Q1 and Q4. This is now clarified as suggested in the revised manuscript:

- table S2: “for the species belonging to each quartile of functional distinctiveness” changed to “for the species belonging to the first (Q1) and last (Q4) quartile of functional distinctiveness”.

- l. 177-178: “We first mapped the spatial distribution of functional distinctiveness, in terms of species richness and total abundance of each functional distinctiveness group” changed to “We first mapped the spatial distribution of functional distinctiveness, in terms of species richness and total abundance of the species belonging to the functional distinctiveness quartiles Q1 (most common) and Q4 (most distinct)”.

- l. 186-190: “For the first RDA, we used the mean total abundance of each functional distinctiveness group and mean values of environmental variables and fishing pressure per ICES rectangle” changed to “For the first RDA, we used the mean total abundance of Q1 and Q4 groups and mean values of environmental variables and fishing pressure per ICES rectangle”.

- l. 190-192: “For the second RDA, we used the annual total abundance of each functional distinctiveness group for the entire North Sea and mean annual environmental variables and fishing pressure” is now replaced by “For the second RDA, we used the annual total abundance of Q1 and Q4 groups for the entire North Sea and mean annual environmental variables and fishing pressure”.

- l. 192-195: For the third RDA, temporal trends of each functional distinctiveness group” replaced by “For the third RDA, temporal trends of the most common (Q1) and most distinct (Q4) species”.

- Fig. 3 title: all mentions of “each functional distinctiveness group” are now replaced by “Q1 and Q4 groups”.

If only Q1 and Q4 were used as response variables, I wonder if it would be more straightforward to run two sets of phylogenetic GLMs (one with Q1 as response and the other with Q4 as response).

We agree with this suggestion: it would have been relevant to apply simple GLMs for these analyses but the advantage of applying the canonical analysis was that it provides biplots, which makes the manuscript easier to read (i.e., through RDA biplots instead of text).

Last, there needs to be more discussion on how warming (after examining the association between temporal trend in distinctiveness vs warming trend and trend in fishing pressure) may be affecting distinctiveness. Presently, there is a good discussion on the effect of fishing pressure but the discussion on warming effects is currently lacking.

Thank you for this relevant point. Additional discussion on warming effects is now included in the revised version. See l. 330 to 337 .

Specific comments

L95. Can the authors provide a more specific URL? The present URL links to a general landing page and does not point to where the IBTS data were downloaded.

A more specific URL is now given in the revised manuscript:

https://datras.ices.dk/Data_products/Download/Download_Data_public.aspx

L111. I suggest adding a brief description of the analysis performed to examine differences in mean distinctiveness across ecological traits [e.g., different habitat affinities, trophic groups, and parental care categories (the analyses associated with Fig. S1)].

We added a sentence in section 2.3 (L. 122-123) to describe the tests performed : “Spearman correlation tests were performed to assess the relationship between each continuous trait and species functional distinctiveness. For categorical traits, differences in functional distinctiveness between trait modalities were tested using Wilcoxon post-hoc tests.”

L120. I suggest changing to “We used Gower distance as a dissimilarity measure because our analyses used both continuous and categorical traits.”

The sentence was modified as suggested.

L133. It is unclear what is the difference between the first analysis “...compare the proportion of IUCN statuses within each functional distinctiveness group” and the second analysis “...compare the proportion of IUCN status between distinctiveness groups”. In the results section 3.2, I understand the analysis as examining how distinctiveness values differed among different IUCN groups. A more straightforward and appropriate analysis will be to use

In this sentence, the first analysis has been removed since the proportion of IUCN categories have been compared only between distinctiveness groups (not within). The sentence has been modified for clarity as follows: “We performed Chi² tests of homogeneity to compare the proportion of IUCN statuses between distinctiveness groups.” (L 133-134). In addition, the figure S3 (boxplot showing distinctiveness of the different IUCN categories) is now added.

L138. I suggest the authors expand on the description of the species vulnerability metric. Please state the source of the data (L131-143) used to calculate the metric.

Cheung et al. calculated and published an index of fishing vulnerability for nearly all fish species listed in FishBase (the largest online data source for fish species information). We obtained published fishing vulnerability scores for each species directly from FishBase using the rfishbase R package. This has now been indicated in the methods (L. 142-144).

L155. The link to the AMO data is broken. Please provide a new updated link to the data.

The link is now corrected : <https://www.psl.noaa.gov/data/timeseries/AMO/>

L163. Should this be updated to be named the “Continuous Plankton Recorder Survey”? And perhaps, this link is better: <https://www.cprsurvey.org/>

This is now corrected in the revised manuscript: “All PCI values were extracted from the Continuous Plankton Recorder Survey dataset (<https://www.cprsurvey.org/>)”.

L177. It is unclear whether the RDAs used all four quartiles of distinctiveness or only Q1 and Q4 as suggested by Fig. 3 and the results section at L235-236. Please clarify. I am guessing that it is the latter (only Q1 and Q4 as suggested in L123-124). If so, I suggest the authors clarify this by revising the language in Table S2: e.g., by changing “for the species belonging to each quartile of functional

distinctiveness” to “ for the species belonging to the first (Q1) and last (Q4) quartile of functional distinctiveness”. Also do revise sentences like the one on L182: e.g., change “each functional distinctiveness group” to “functional distinctiveness quartiles Q1 (most common) and Q4 (most distinct)” and others e.g., the one on L187.

This is now clarified in the revised manuscript (see response to comment above).

L207. Insert a full stop after “spp”.

The full stop is now included.

L210. Typo error. It should be “primarily’ instead of “primary”.

This was corrected accordingly.

L216. Provide the results of the chi-square test for homogeneity.

The statistics were added as suggested: “We found no significant link between species functional distinctiveness and IUCN status (Chi² test; p-value = 0.682; Figure S3)” (l. 217).

L229. Suggest inserting “more” before “functionally common species”.

In order to keep the same formulation throughout the entire manuscript, we used “functionally common species” to designate Q1 species. The word/information given by the word ‘more’ is given l. 125-126: “The first group (Q1) included the most functionally common species while the fourth group (Q4) included the most functionally distinct species”.

Fig. 1. Figure 1 contains information regarding IUCN status, vulnerability to fishing, and functional distinctiveness, which are the three variables relevant to the analysis, which provides a good level of information to readers. Specifically, the figure plots functional distinctiveness against fishing vulnerability. I suggest a further improvement to the figure so that it provides information that is more directly relevant to the first analysis: the relationship between functional distinctiveness and IUCN status. To do that, I suggest adding an inset figure of boxplots of distinctiveness values in different IUCN categories (i.e., like the boxplots in Figure S1).

Thank you for your suggestion. We didn’t succeed in adding the boxplot in the current figure 1 but it is now included in the supplementary material (Figure S3) and we refer to it in the main text (l. 229-231).

Table S3. Suggest changing “Reparation” to “List”.

This was changed as suggested.

L235. How about the trends in the Q2 and Q3 species? Did the abundance of Q2 and Q3 species also rebound?

Q2 and Q3 species also increased over the last 30 years in the North Sea. For better understanding and because we wanted to emphasize this work on the concept of rarity, we only selected Q1 and Q4 and then only focused on the most distinct and common species.

Fig. 3. Interesting that the dropoff in the variance explained by RDA2 (2-8%) compared to RDA1 (44-67%) was so great. This has implications for the interpretation and inference: that it is more important to look at the trends and associations with respect to the x-axis (RDA1).

Yes, because of that dropoff, the first axis was the one we essentially used to interpret our results.

Fig. 3. I also suggest adding points (of each observation) to the RDA plots, i.e., make it a triplot. I understand that it may make the figure a little messier and busier, but including the observations [individual ICE rectangles in (a) and (c) with color gradient reflecting latitude; individual years in (b) with color gradient reflecting year] can help the reader interpret the gradients and associations with environmental variables and functional distinctiveness groups better.

Points are now included in the three RDAs (Figure 3), with colour gradients.

Also, did the RDA plots use Type 1 or Type 2 scaling? To the best of my knowledge, Type 2 scaling is appropriate when the aim is to illustrate associations between the response variables (i.e. relationships between Q1 and Q4 abundances), while Type 1 scaling is appropriate when the aim is to illustrate distances or similarities among observations. One option is to present Type 1 scaling in the full text, since the authors want to discuss the lat and time gradient; and present Type 2 scaling in the SI, where the focus can be on the correlation between Q1 and Q4 responses.

RDAs were performed with scaling = 2. That scaling is used in order to investigate correlations between distinctiveness groups and driving forces. This information (i.e., 'scaling = 2' is now included in the 3 RDAs of the Fig. 3).

L233. According to Fig. 3a, PCI is also important as it has a highly negative RDA1 value (only slightly different from SST).

Phytoplankton Colour Index (PCI) was added to the list of the variables that most explained the spatial RDA (Figure 3a) (L. 239-242): "In the spatial RDA (question 1 in Table S2), 52% of variance was explained by environmental variables and fishing pressure, with depth, sea surface temperature (SST), Phytoplankton Colour Index (PCI) and trawling effort being the variables that best explained the spatial distribution of functional distinctiveness group abundance (Figure 3a)."

L261. "Functional distinctiveness was highly related to particular demographic strategies as well as habitat preferences" – This point is a given because functional distinctiveness is based on these traits and habitat affinities so distinctiveness has to be based on them. I rephrasing the sentence by replacing "was highly related to" with "was contributed by".

This was changed to "Functional distinctiveness was shaped by particular demographic strategies..." (L. 268-271).

L276. Suggest replacing "support that" with "indicate that".

This was changed as suggested.

L282. I don't think the authors need to cite the figure because this is a result that has already been presented in the results section.

In fact, the reason for which this figure is cited here is because all supplementary figures must be referenced in the main text, and this figure had not been previously referenced in the paper.

L228-231 and L279-282. Suggest tightening up the sentence here. What does more frequent mean? Importantly, looking at Fig. S2, there is not a clear gradient in the species richness of functionally distinct species from north to south (perhaps a very slight gradient; Fig. S2b), and for dominance (Fig.

S2d), the pixels of highest dominance seem to be in middle latitudes between the north and south. Same pattern is observed with abundance and numerical dominance (Fig. S2f and S2h respectively). I think the authors need to acknowledge that the N-S gradient for Q4 species isn't as strong as the gradient for functionally common species.

Thank you for pointing this out. This is now modified as suggested. In addition the expression 'more frequent' is now reworded:

L234-239: "The species richness and total abundance of functionally distinct species were higher in deeper, colder, and more saline environments (i.e., northern North Sea), while functionally common species were found in shallower, warmer environments with higher phytoplankton biomass, trawling effort and bed shear stress (i.e., southern North Sea; Figures 2 and S4). However, it should be noted that the latitudinal gradient for functionally distinct species (Q4) was less pronounced than for functionally common species (Q1)".

L287-291: "Although less pronounced for functionally distinct species, we found an opposite latitudinal gradient in the distribution of functionally distinct vs. functionally common species in the North Sea. Functionally distinct species were especially found in the north (i.e., highest Q4 species richness; Figure S4) while functionally common species, with a clearer latitudinal gradient, were especially located in the southern North Sea (i.e., higher Q1 species richness and abundance; Figure S4).

L297-282. I think the manuscript is missing a discussion on the mechanisms by which climate warming may have affected the spatiotemporal dynamics of functional distinctiveness. The emphasis was on fishing pressure here, but I think readers would be interested to read about how warming may also be important, given that warming was in the title of the paper.

This is now taken into account (see our response to one of your main comments above).

L288-296. I think the two hypotheses here need to be supported by previous studies, if possible. Also, the paper will benefit from a brief discussion on the potential mechanisms behind the association between r-K gradient, and gradient of seasonality/deeper waters (environmental stability).

Further discussion on the mechanisms behind the association between r-K gradient, and environmental stability is now included, in addition to several references (l. 306-309 and l. 330-337).

L297-307. I am not sure if the data and analyses support the idea that reduced fishing is particularly important for the rebound in functionally distinct species particularly in the southern North Sea. Looking at the second row of panels in Fig. 2, I think the abundance of Q4 species has increased in both northern and southern regions. It is not particularly apparent that the rebound in the south is more pronounced than that in the north, perhaps by a little bit. Also in Fig. 3, Q4 is positively related to "trawl", which suggests that sites with higher "trawl" values (i.e., higher fishing pressure), are also the sites where the abundance of functionally distinct species was increasing with time.

Thank you again for that important comment. The RDA indicates that functionally distinct species (Q4) mostly increased in abundance where fishing pressure was historically higher (i.e., in the south). In all cases we agree with you that the latitudinal gradient in Q4 rebound exists but is not so pronounced (by looking at the fig 2). For that reason, we reworded all sentences related to that spatial gradient of Q4 temporal trends (by toning down the latitudinal gradient) in l. 28-29, 313-314.

L363. Typo error. It should be "Funding" rather than "Founding".

This has been corrected.